# OpenReview forum: "CNFINBENCH: A BENCHMARK FOR SAFETY AND COMPLIANCE OF LARGE LANGUAGE MODELS IN FINANCE"
_ICLR.cc/2026/Conference — ICLR 2026 Conference Withdrawn Submission_

### Official Review · Reviewer_z2j2 · 2025-10-28

**Soundness:** 3
**Presentation:** 2
**Contribution:** 4
**Rating:** 6
**Confidence:** 3

**Summary:**

The paper introduces CNFinBench, the first large-scale benchmark specifically designed to evaluate the safety, compliance, and capability of LLMs in financial applications. It goes beyond traditional factual QA benchmarks by testing whether models behave responsibly under realistic and adversarial financial scenarios. The benchmark includes over 13,000 tasks and multi-turn dialogues, covering regulatory compliance, risk control, and ethical financial advising. The data construction goes through multiple rounds of filtering and quality checking from the combination of LLMs and human experts. Evaluation is performed adaptively accordingly to the task type, using both expert review and LLM ensemble judging to ensure scalability.
The evaluation is extensive and comprehensive, involving a large number of commercial and open-source LLMs.

**Strengths:**

- First benchmark focusing on financial safety and regulatory compliance.
- Incorporates adversarial dialogues and jailbreak stress tests.
- Combines expert review with LLM ensemble judging for scalable evaluation.
- Extensive evaluation across many models.

**Weaknesses:**

- Potential Data Bias.
- Main results analysis could be more detailed.
- LLM Consensus as a Confidence Signal not investigated.

**Questions:**

Overally, this paper presents a significant contribution to the evaluation of LLMs in the financial domain, addressing critical aspects of safety and compliance that are often overlooked in general benchmarks. The CNFinBench benchmark is well-designed, covering a wide range of realistic scenarios and incorporating both human and automated evaluation methods. The extensive experiments provide valuable insights into the performance of various LLMs in this specialized context. However, there are several areas where the paper could be improved:

**(1) Potential Data Bias.**

In Line 225-234, the authors mention that 70% of the dataset is constructed (filtered) using advanced models (e.g., GPT-4o, Qwen3, DeepSeek-R1, Gemini-2.5).
And in the main result, for example, Figure 4, these models often outperform others. This raises concerns about potential data bias, as models used to generate or filter the dataset may have an inherent advantage in evaluation.
I recommend the authors to conduct additional analyses to assess the extent of this bias. For instance, they could evaluate model performance on a subset of the dataset that was not influenced by these advanced models. This would help determine whether the observed performance differences are due to genuine model capabilities or artifacts of the dataset construction process.

**(2) The analysis of the main results could be more detailed.**

In Table 1 and 2, there are more interesting observations that could be discussed.
For example, it is observed that qwen3-23B performs better than qwen3-32B in the capability tasks. This is counterintuitive, as larger models typically perform better. The authors could explore potential reasons for this anomaly.
What's more, where reasoning models (e.g., DeepSeek) and Instruction-tuned models would perform better in these tasks?
Whether involving CoT during inference would help improve the performance?

**(3) Investigate LLM Consensus as a Confidence Signal.**

Appendix E, the results are in Table 7 (not Table 6).
The paper validates LLM judges against human experts but doesn't analyze when the three LLMs agree with each other. I recommend investigating whether unanimous LLM agreement correlates with correctness. Specifically, measure the accuracy when all three judges give the same score versus when they disagree. If unanimous cases show significantly higher accuracy, this could enable a practical mixed strategy: auto-accept high-consensus cases while reserving human review for disagreements. This would be especially valuable for extending the dataset, as it could substantially reduce human validation costs while maintaining quality.
In addition, some tasks are easier to judge (Capability: κ=0.81) while others are harder (Safety: κ=0.68). Probably use adaptive rules for harder tasks.

**Minor:**

What is the content in Appendix I in Line 1830-1831?

---

### Official Review · Reviewer_imuq · 2025-11-01

**Soundness:** 3
**Presentation:** 3
**Contribution:** 3
**Rating:** 4
**Confidence:** 3

**Summary:**

This paper introduces CNFinBench, a large-scale benchmark designed to evaluate large language models (LLMs) in financial domains with a focus on safety, compliance, and capability. Unlike prior finance-oriented benchmarks, which primarily test factual and numerical proficiency, CNFinBench explicitly measures regulatory adherence, ethical conduct, and adversarial robustness.

The benchmark comprises over 13,000 single-turn and 100 multi-turn adversarial instances across 15 subtasks grouped into three families: Safety tasks (e.g., jailbreak and prompt-injection resistance); Compliance and risk-control tasks (e.g., KYC/AML, disclosure integrity); Capability tasks (financial QA, reasoning, report parsing, text generation).

LLM judging ensemble (GPT-4o, Gemini-2.5-Pro, and DeepSeek-V3/Qwen3-235B) is used for scalable scoring, with expert calibration to verify alignment. The authors show that even high-performing general models (e.g., GPT-5) display a capability–compliance gap—performing well on structured finance tasks but failing on safety or ethical dimensions.

**Strengths:**

* The benchmark introduces nine finance-specific red-team attack categories (e.g., fallacy, topic-drift, obfuscation, chained questioning) and dynamic option perturbation for robustness testing.

* The dataset is constructed via expert-AI collaborative verification, Delphi-style rounds with 210 finance professionals, and iterative filtering with multiple frontier LLMs to remove trivial items

* CNFinBench fills a critical evaluation gap for high-risk AI systems under the EU AI Act and similar regulations.

**Weaknesses:**

* Despite attempts to minimize overlap between judges and tested systems, shared training data among LLMs might inflate correlation scores.

* Heavy reliance on synthetic data and LLM-generated content. This risks style leakage—models tested on CNFinBench may benefit from exposure to similar language patterns or red-team styles during pretraining, inflating scores.

* The benchmark claims to span “finance-specific capabilities” but the taxonomy mixes heterogeneous abstraction levels. These tasks are not orthogonal; for example, a single failure in compliance reasoning could penalize a model for both capability and safety dimensions. The design lacks task-level independence validation—no correlation or factor analysis is presented to show that each subtask measures a distinct construct. Consequently, aggregate metrics (e.g., “capability–compliance gap”) may conflate construct overlap with genuine performance differences.

* CNFinBench is static, containing 2024–2025 filings and risk cases. Financial regulation evolves rapidly (e.g., Basel IV, MiFID amendments), so benchmark validity decays quickly. There is no mechanism for continuous updating or version control, which undermines long-term reproducibility and comparability of results across years.

**Questions:**

* The benchmark spans 15 subtasks, nine adversarial categories, and multi-turn dialogues, yet the integration logic across tasks is unclear—how scores from diverse modalities (MCQ, open QA, generation, dialogue) are normalized and aggregated into a unified evaluation metric?

* Even though human–LLM agreement is reported via Cohen’s κ, inter-judge calibration and disagreement resolution are not disclosed—did the authors use majority voting, confidence weighting, or rubric aggregation?

---

### Official Review · Reviewer_Wvca · 2025-11-01

**Soundness:** 2
**Presentation:** 1
**Contribution:** 2
**Rating:** 2
**Confidence:** 4

**Summary:**

The paper proposes CNFinBench, a benchmark for evaluating LLMs in finance, emphasizing safety, compliance, and capabilities. It categorizes tasks into Safety (adversarial jailbreaks, multi-turn dialogues), Compliance/Risk Control (regulatory auditing, risk assessment), and Capabilities (QA, analysis, computation). Innovations include finance-specific attacks (e.g., nine jailbreak strategies), evidence-grounded long-document tasks, and a three-LLM judge ensemble with human calibration. Experiments on 23 models reveal capability-compliance gaps, aiming to support safe LLM deployment in finance.

**Strengths:**

1. Incorporates authentic sources like 400 A-share reports and regulations (CBIRC, IFRS, SEC), with expert validation (210 for taxonomy, 21 for anti-fraud), enhancing domain relevance.
2. Multi-turn dialogues and tailored jailbreaks test dynamic safety effectively.
3. LLM ensemble judging with high human agreement.

**Weaknesses:**

1. Relies on existing paradigms from general benchmarks and finance ones, lacking new ML techniques or theoretical contributions to representation learning.
2. No ablations (e.g., judge sensitivity), statistical tests, or non-LLM baselines; high variance without error analysis.
3. Benchmark/code not confirmed public; 70% LLM-generated data risks biases.
4. Overstated claims, like "Holistic evaluation" ignore gaps in rigor, e.g., no causal insights into gaps.

**Questions:**

1. Why no ablations on judge ensemble?
2. How were jailbreaks validated as finance-unique?

---

### Official Review · Reviewer_r8zQ · 2025-11-06

**Soundness:** 3
**Presentation:** 1
**Contribution:** 2
**Rating:** 4
**Confidence:** 3

**Summary:**

The paper introduces a evaluation framework designed to assess LLMs used in financial applications. The authors propose CNFinBench, a benchmark that measures model performance across three major domains: Safety, Compliance and Risk Control, and Capability. The benchmark contains over 13,000 instances and 100 multi-turn adversarial dialogues, covering regulatory compliance, ethical behavior, and financial reasoning. It incorporates nine categories of adversarial attacks, dynamic option perturbation for objective tests, and a three-model judging ensemble calibrated with human experts. Experiments across 21 models reveal a notable gap between technical capability and compliance robustness: models strong in financial reasoning often underperform in safety and regulatory adherence. CNFinBench thus provides a reproducible and auditable framework for evaluating and aligning financial LLMs toward safer, regulation-aware performance

**Strengths:**

1. Introduces a benchmark specifically focused on safety and regulatory compliance in financial LLMs.
2. Incorporates human financial experts throughout dataset design, validation, and calibration of evaluation results.

**Weaknesses:**

1. **Insufficient methodological explanation:**
   * Section 3 lists numerous regulations and standards but provides explanation of how these sources inform dataset construction.
   * The proposed benchmark pipeline (Figure 2) outlines a multi-step process, but each step lacks sufficient elaboration. This section should form the methodological core of the paper but remains underdeveloped.
   * Despite extensive discussion of the benchmark, the paper does not present any concrete data instances or illustrative examples. Without sample items, it is difficult to grasp the dataset.

2. **Unrealistic experimental setup:**
   The experimental configuration, which restricts generation length to 512 tokens, context window to 2k tokens, and excludes tool use or external data access, does not reflect real-world financial applications. Such constraints likely suppress realistic model behaviors, especially those related to sensitive information leakage or misuse. Without incorporating at least mock sensitive data or simulated function calls, it is unclear how harmful behaviors could manifest in the safety evaluations.

3. **Unclear design of multi-turn dialogue simulation:**
   The paper claims to include multi-turn dialogues for safety evaluation, but it does not explain how these dialogues are generated or simulated. A realistic conversation requires modeling both the attacker and the assistant sides, yet the paper only presents final dialogues without clarifying how they were created. If dialogues are prewritten rather than dynamically generated by models, it is debatable whether they constitute genuine “multi-turn simulations.”

### **Minor Issues**

1. In Figure 1, “Exper-AI” should be corrected to “Expert-AI.”
2. The dialogue text in Figure 3 is difficult to read due to poor color contrast.

**Questions:**

1. What is meant by the “Delphi round” in Figure 2?
2. How is the dataset synthesized in practice? Could the authors provide examples of synthesis prompts or templates?

---

### Note · Authors · 2026-02-24

I have read and agree with the venue's withdrawal policy on behalf of myself and my co-authors.

---

### Meta-Review · Area_Chair_VaX4 · 2025-12-30

**Summary:**

The paper established CNFinBench, a new benchmark to evaluate the safety and compliance of large language models in finance. The benchmark contains over 13,000 instances and 100 multi-turn adversarial dialogues, covering regulatory compliance, ethical behavior, and financial reasoning. Experiments on diverse models reveal the capability-compliance trade-off.

The reviewers have found some strengths of this paper, including a new benchmark focused on finance LLM, incorporating adversarial testing and other techniques for comprehensive evaluation, considering human evaluation, etc. However, the reviewers also raised several major concerns about this paper.

- Insufficient methodological details: Some reviewers found that the dataset construction and benchmark pipeline are not explained clearly, limiting reproducibility and interpretability.
- Potential data and evaluation bias: Some reviewers argued that the benchmark is heavily relied on LLM-generated data, risking style leakage for models involved in data creation or judging, with no ablations to quantify this effect.
- Unrealistic evaluation setup: The authors put strict constraints on context length, generation, and tool use, which does not reflect real financial applications.
- Limited statistical analysis: The study lacks ablations, error analysis, statistical testing, non-LLM baselines, and sometimes overstates claims such as “holistic evaluation.”

**Reviewer Concerns:**

The authors did not submit rebuttal. Thus I believe the reviewer concerns are not addressed.

**Reviewer Scores:**

Since no author rebuttal is provided, I think the reviewer scores would not change.

---

### Decision · Program_Chairs · 2026-01-26

Reject